

# The association between *ADIPOQ* gene variants (rs266729, rs2241766, rs1501299) and acute myocardial infarction in Vietnamese patients with type 2 diabetes mellitus

Bao Hoang Le[1], Khanh Quang Tran[2,3], Nhu Nhat Quynh Nguyen[4] and Truc Thanh Thai[5]

[1] Department of Endocrinology, University Medical Center at Ho Chi Minh City, Ho Chi Minh City, Vietnam
[2] Department of Endocrinology, University of Medicine and Pharmacy at Ho Chi Minh City, Ho Chi Minh City, Vietnam
[3] Department of Endocrinology, Nguyen Tri Phuong Hospital, Ho Chi Minh City, Vietnam
[4] Center for Molecular Biomedicine, University of Medicine and Pharmacy at Ho Chi Minh City, Ho Chi Minh City, Vietnam
[5] Department of Medical Statistics and Informatics, University of Medicine and Pharmacy at Ho Chi Minh City, Ho Chi Minh City, Vietnam

Corresponding author
Truc Thanh Thai,
thaithanhtruc@ump.edu.vn

## ABSTRACT

**Background:** Type 2 diabetes mellitus (T2DM) significantly increases the risk of acute myocardial infarction (AMI). Polymorphisms in the ADIPOQ gene, which encodes the adiponectin hormone, are believed to influence cardiovascular disease risk. This study aims to evaluate the association between three ADIPOQ gene variants—rs266729, rs2241766, and rs1501299—and the risk of AMI in Vietnamese patients with T2DM.

**Methods:** A case–control study was conducted from January 2023 to June 2024 at the University Medical Center, Ho Chi Minh City. The study included 275 T2DM patients with a history of AMI (case group) and 275 T2DM patients without AMI (control group). Participants were matched by age, gender, smoking status, and duration of T2DM to control for potential confounding factors. DNA was extracted from peripheral blood samples, and the three polymorphisms were genotyped using Sanger sequencing.

**Results:** Significant associations were found between ADIPOQ gene polymorphisms and AMI risk. The G/G genotype of rs266729 and the T/T genotype of rs1501299 were associated with a reduced risk of AMI (OR = 0.45, 95% CI [0.23–0.89], $p = 0.021$; OR = 0.45, 95% CI [0.22–0.90], $p = 0.025$, respectively). In contrast, the T/G genotype of rs2241766 was significantly associated with an increased risk of AMI (OR = 2.07, 95% CI [1.44–2.97], $p < 0.001$).

**Conclusion:** This study suggests that rs266729 and rs1501299 may have protective effects, while rs2241766 may increase the risk of AMI in Vietnamese patients with T2DM. These findings highlight the importance of further research into adiponectin

levels and long-term monitoring, and support the potential use of genetic variants in personalized cardiovascular risk management for patients with T2DM.

## INTRODUCTION

Type 2 diabetes mellitus (T2DM) is a major risk for cardiovascular diseases, particularly acute myocardial infarction (AMI) (*Kannel & McGee, 1979*; *Milazzo et al., 2021*). T2DM patients face higher AMI risk, mortality, and post-AMI complications. A systematic review and meta-analysis (*Gholap et al., 2017*) of over 700,000 patients showed 50% higher long-term post-AMI mortality, independent of AMI phenotype or modern treatment.

Though T2DM is incurable, investigating AMI risk factors in T2DM patients is crucial for optimal treatment. Numerous studies have been conducted to identify AMI risk factors, but they have primarily focused on traditional clinical and subclinical factors. The advances in medicine and genetic technologies have opened new avenues for research on potential risk factors, including the *ADIPOQ* gene, which has been reported in previous studies (*Hou et al., 2017*). The *ADIPOQ* gene encodes adiponectin, an adipokine vital for glucose and lipid metabolism and cardiovascular protection. Several polymorphisms in this gene, particularly rs266729, rs2241766, and rs1501299, have been widely studied due to their influence on circulating adiponectin levels and the associated cardiovascular risk. Studies have shown that changes in these polymorphisms can reduce adiponectin levels, thereby increasing insulin resistance, atherosclerosis, and AMI risk in T2DM patients (*Antonopoulos et al., 2013*).

Although research across various populations worldwide has examined the association between *ADIPOQ* gene polymorphisms and AMI risk in T2DM patients, results remain inconsistent. For example, *Antonopoulos et al. (2013)* found rs1501299 T/T increased AMI risk threefold in Greeks. In contrast, *Zhong et al. (2010)* found no significant association for rs822395 or rs266729 with coronary artery disease/acute myocardial infarction (CAD/AMI) in the Chinese population. Similarly, *Filippi et al. (2005)* (Italian) found no clear association between the rs2241766 variant and CAD risk, although noted lower adiponectin levels. Conversely, *Ji et al. (2018)* reported rs1501299 protective effects against AMI in Korean males. Research by *Rizk et al. (2013)* on Egyptian patients also noted interactions between *ADIPOQ* gene variants and risk factors such as insulin resistance, but no clear association with AMI was identified. These disparities suggest that *ADIPOQ* variant effects depend on population-specific genetic/environmental factors, highlighting the need for more studies.

In Vietnam, T2DM and AMI remain major health burdens, with an estimated 4 million T2DM patients aged 20–79 years (*Sun et al., 2022*), and T2DM linked to 21% of AMI hospitalizations (*Nguyen et al., 2023*). However, limited local research exists on *ADIPOQ* gene polymorphisms and AMI risk in Vietnamese T2DM patients. Such studies are crucial

to clarify the *ADIPOQ* gene variants-AMI link and inform prevention/treatment strategies. Therefore, this study aims to evaluate how the rs266729, rs2241766, and rs1501299 variations in the *ADIPOQ* gene are related to the risk of acute myocardial infarction in patients with type 2 diabetes in Vietnam.

## MATERIALS AND METHODS

### Study setting and design
A case-control study was conducted from March 2023 to June 2024 at the University Medical Center in Ho Chi Minh City.

### Study population
Study participants were selected by a consecutive sampling method, including patients aged 18 years and older and diagnosed with T2DM in two groups. The case group was T2DM patients hospitalized for AMI and treated at the Interventional Cardiology Department. The control group was T2DM patients with no history of AMI, clinically stable with a normal resting electrocardiogram (ECG), being followed up and treated at the Endocrinology Department. Patients in the control group were matched with those in the case group based on age, gender, smoking status, and duration of T2DM. Non-Kinh ethnicities and pregnant women were excluded from the study.

The sample size was calculated according to the guidelines by S. Purcell and P. Sham for individual genetic traits in case-control studies (*Purcell, Cherny & Sham, 2003*). Based on the high-risk allele frequency for each single nucleotide polymorphism (SNP) in the *ADIPOQ* gene, the minimum total sample size was 550, including 275 T2DM patients with AMI (case group) and 275 T2DM patients without AMI (control group).

### Measurements
AMI was defined according to the 2018 ESC/ACC/AHA/WHF criteria (*Thygesen et al., 2018*), which required a rise and/or fall of cardiac troponin values, with at least one value above the 99th percentile upper reference limit, accompanied by at least one of the following: (1) symptoms of myocardial ischemia; (2) new ischemic ECG changes (ST-T changes or new left bundle branch block); (3) development of new pathological Q waves on the ECG; (4) new regional wall motion abnormalities on echocardiography; or (5) identification of intracoronary thrombus by angiography.

The primary independent variables were the *ADIPOQ* gene variants, including their respective genotypes (*e.g.*, C/C, C/G, G/G for rs266729 where C is the reference allele; T/T, T/G, G/G for rs2241766 where T is the reference allele; G/G, G/T, T/T for rs1501299 where G is the reference allele), identified from peripheral blood samples using Sanger sequencing. Peripheral blood samples (2 mL) were collected in EDTA tubes and stored at 4 °C in 2 days before long-term preservation at −80 °C for DNA analysis.

*DNA extraction*: Genomic DNA was extracted from blood cells using the GeneJET Whole Blood Genomic DNA Purification Mini Kit (#K0782; Thermo Fisher Scientific, Waltham, MA, USA). Following the manufacturer's instructions, 200 μL of the genomic DNA

solution (gDNA) was obtained. The purity and concentration of the gDNA were measured using the NanoDrop2000 Spectrophotometers (#ND-2000, Thermo Fisher Scientific, Waltham, MA, USA). The sample with ratio of A260/280 nm from 1.8 is accepted as high purity DNA and was stored at −20 °C for further analysis.

*Polymerase chain reaction and primer design*: Polymerase chain reaction (PCR) primers were designed based on the standard sequence of the *ADIPOQ* gene from GenBank using CLC Main Workbench v5.5 software. The primers specifically targeted the rs1501299, rs2241766, and rs266729 polymorphisms. The PCR reactions were prepared with 50 ng of DNA and a final master mix for each reaction containing 1X PCR Buffer (#R007A; Takara Bio, Kusatsu, Shiga, Japan), 200 μM dNTP (#R007A; Takara Bio, Kusatsu, Shiga, Japan), 0.5 U of Takara Taq™ Hot Start Polymerase (#R007A; Takara Bio, Kusatsu, Shiga, Japan), and 0.5 μM each of forward and reverse primers. Amplification was carried out with an initial denaturation at 98 °C for 3 min, followed by 32 cycles (98 °C for 10 s, 58 °C for 20 s, and 72 °C for 50 s), and a final extension step at 72 °C for 2 min. In addition, a no-template control containing nuclease-free water was included in each PCR run to monitor potential DNA contamination. The PCR products were checked using 2% agarose gel electrophoresis (UVP-GelDoc-It 310 Imaging System, Thermo Fisher Scientific, Waltham, MA, USA) with a 1 kb plus DNA ladder (#10787018; Invitrogen, Waltham, MA, USA) to confirm the expected fragment size.

*PCR product purification*: PCR products were purified using the ExoSAP-IT® PCR Product Cleanup kit (#78201.1.1ML; Applied Biosystems, Waltham, MA, USA), which included Exonuclease I and Shrimp Alkaline Phosphatase to remove unwanted products. The purified PCR products were used for the cycle sequencing reaction with the BigDye® Terminators v3.1 kit (#4337455; Thermo Fisher Scientific, Waltham, MA, USA).

*Gene sequencing and result analysis*: Sequencing was performed using an ABI 3500 Genetic Analyzer (#4405673; Applied Biosystems, Waltham, MA, USA). The nucleotide sequences were analyzed by comparing them to reference sequences (NG_021140) from NCBI using CLC Main Workbench v5.5, identifying genotype changes at the SNPs rs1501299, rs2241766, and rs266729 of the *ADIPOQ* gene.

In addition, other patient's characteristics were collected, including baseline demographics (age, sex, duration of T2DM, family history of T2DM), comorbidities (hypertension, dyslipidemia, stroke, peripheral artery disease), lifestyle habits (smoking, physical activity), body mass index (BMI), waist-to-hip ratio, blood pressure, lipid profile, blood glucose, HbA1c, renal function, ejection fraction (measured by Simpson's method), and current medications. Laboratory tests for glucose, HbA1c, total cholesterol, LDL-C, HDL-C, triglycerides, and creatinine were also recorded.

## Statistical analysis

Data were analyzed using Stata 17.0 software. Categorical variables were presented as frequencies and percentages, while continuous variables were presented as means and standard deviation or medians and interquartile ranges for non-normally distributed data.

For continuous variables, data normality was assessed using visual inspection of histograms and the Shapiro-Wilk test. Prior to association analysis, the genotype frequencies of all three SNPs (rs266729, rs2241766, and rs1501299) in the control group were tested for Hardy-Weinberg equilibrium using a Chi-squared test. The Chi-squared test was used to compare proportions, while the t-test and Mann-Whitney U test were used to compare means between the two groups. Univariate and multiple logistic regression were used to assess the association between genotypes and AMI, with and without adjustment for other clinical and subclinical factors. The effect sizes were reported as Odds Ratios (ORs) with their 95% confidence intervals. The significance level ($\alpha$) was set at 0.05; therefore, a $p$-value $< 0.05$ was considered statistically significant.

## Ethical considerations

The study complied with the ethical guidelines for biomedical research and was approved by the Biomedical Research Ethics Committee of the University of Medicine and Pharmacy at Ho Chi Minh City (Approval number: 183/HDDD-DHYD, dated: 16/02/2023). All participants were fully informed about the study's objectives, procedures, and their rights, and they provided written informed consent. Personal information of patients was kept strictly confidential.

## RESULTS

Table 1 presents the sociodemographic characteristics and medical history of T2DM patients included in the study. The case and control groups were well-matched for age, gender, smoking status, and duration of T2DM ($p > 0.05$ for all). Specifically, the age of patients in this study ranged from 31 to 94 years. The median duration of T2DM in the case and control groups was 6.0 years (IQR 1.0–15.0) and 6.0 years (IQR 2.0–14.0), respectively. However, as expected, the AMI case group exhibited a significantly higher burden of cardiovascular comorbidities, including hypertension, stroke, and peripheral artery disease ($p < 0.01$ for all). Moreover, the case group demonstrated a more adverse cardiometabolic profile, characterized by poorer glycemic control, dyslipidemia, and impaired renal function (eGFR) ($p < 0.001$ for all) (Table 2). All these significant factors were subsequently adjusted for in the multivariate regression models.

Table 3 summarizes and compares the distribution of genotypes, alleles, and haplotypes of the three studied SNPs (rs266729, rs2241766, rs1501299) between the two groups. For rs266729 (−11,377 C/G), the G/G genotype was less frequent in the case group than in the control group (5.1% $vs.$ 10.5%, $p = 0.058$). The G allele was less frequent among AMI patients, though this difference was not statistically significant ($p = 0.092$). For rs2241766 (+45 T/G), the T/G genotype was significantly more frequent in the case group (45.1%) compared to the control group (29.5%, $p < 0.001$). The G allele of rs2241766 was more frequent in the case group (32.7% $vs.$ 24.9%, $p = 0.004$). For rs1501299 (+276 G/T), the T/T genotype was less frequent in the case group (4.7%) compared to the control group (9.5%, $p = 0.069$). The T allele of rs1501299 was less frequent in the case group (22.4% $vs.$ 27.8%, $p = 0.037$). Haplotype analysis revealed that the most common haplotype in both groups was C-T-G, which was used as the reference. The G-T-T haplotype was less frequent in the

**Table 1 Sociodemographic and medical history characteristics of T2DM patients in the study.**

| Characteristic | Groups | | p-value |
|---|---|---|---|
| | Case (n = 275) | Control (n = 275) | |
| Age (years), *mean (SD)* | 67.3 (11.3) | 67.2 (11.5) | 0.905 |
| **Gender** | | | |
| Female | 122 (44.4) | 122 (44.4) | 0.999 |
| Male | 153 (55.6) | 153 (55.6) | |
| **Duration of T2DM (years),** *median (IQR)* | 6.0 (1.0–15.0) | 6.0 (2.0–14.0) | 0.340 |
| **Family history of diabetes** | | | |
| No | 146 (53.1) | 149 (54.2) | 0.798 |
| Yes | 129 (46.9) | 126 (45.8) | |
| **Hypertension** | | | |
| No | 24 (8.7) | 49 (17.8) | 0.002 |
| Yes | 251 (91.3) | 226 (82.2) | |
| **Dyslipidemia** | | | |
| No | 8 (2.9) | 8 (2.9) | 0.999 |
| Yes | 267 (97.1) | 267 (97.1) | |
| **History of stroke** | | | |
| No | 238 (86.5) | 274 (99.6) | <0.001 |
| Yes | 37 (13.5) | 1 (0.4) | |
| **History of peripheral artery disease** | | | |
| No | 232 (84.4) | 271 (98.5) | <0.001 |
| Yes | 43 (15.6) | 4 (1.5) | |
| **Smoking status** | | | |
| No | 181 (65.8) | 181 (65.8) | 0.999 |
| Yes | 94 (34.2) | 94 (34.2) | |
| **Physical activity** | | | |
| No | 118 (42.9) | 126 (45.8) | 0.492 |
| Yes | 157 (57.1) | 149 (54.2) | |

**Table 2 Clinical and laboratory characteristics of T2DM patients in the study.**

| Characteristic | Groups | | p-value |
|---|---|---|---|
| | Case (n = 275) | Control (n = 275) | |
| **Body mass index (kg/m$^2$),** *mean (SD)* | 23.8 (3.4) | 23.7 (2.9) | 0.522 |
| **Waist circumference (cm),** *mean (SD)* | 92.6 (9.9) | 92.2 (9.4) | 0.662 |
| **Hip circumference (cm),** *mean (SD)* | 94.9 (9.6) | 97.6 (9.4) | 0.001 |
| **Waist-to-hip ratio,** *mean (SD)* | 1.0 (0.1) | 0.9 (0.1) | <0.001 |
| **Systolic blood pressure (mmHg),** *mean (SD)* | 122.3 (20.0) | 133.3 (18.3) | <0.001 |
| **Diastolic blood pressure (mmHg),** *mean (SD)* | 72.8 (11.8) | 78.3 (10.5) | <0.001 |
| **Glucose (mg/dL),** *mean (SD)* | 188.6 (76.9) | 147.1 (50.2) | <0.001 |

| Table 2 (continued) | | | |
|---|---|---|---|
| Characteristic | Groups | | p-value |
| | Case (n = 275) | Control (n = 275) | |
| HbA1c (%), *mean (SD)* | 8.4 (1.9) | 7.8 (1.8) | <0.001 |
| Total cholesterol (mg/dL), *mean (SD)* | 175.9 (57.7) | 159.8 (48.0) | 0.001 |
| HDL-C (mg/dL), *mean (SD)* | 40.1 (10.6) | 42.6 (11.5) | 0.010 |
| LDL-C (mg/dL), *mean (SD)* | 115.1 (41.5) | 94.0 (35.7) | <0.001 |
| Triglycerides (mg/dL), *median (IQR)* | 159.5 (119.0–242.5) | 152.0 (111.0–213.0) | 0.042 |
| Creatinine (mg/dL), *mean (SD)* | 1.4 (1.3) | 1.0 (0.5) | <0.001 |
| eGFR (mL/min/1.73m$^2$), *mean (SD)* | 65.5 (26.3) | 78.3 (24.1) | <0.001 |
| Ejection fraction (%), *mean (SD)* | 50.0 (14.9) | 66.0 (6.9) | <0.001 |
| Antidiabetic medications | | | |
| Sulfonylureas | 24 (8.7) | 102 (37.1) | <0.001 |
| Metformin | 50 (18.2) | 199 (72.4) | <0.001 |
| DPP-4 inhibitors | 59 (21.5) | 190 (69.1) | <0.001 |
| SGLT2 inhibitors | 183 (66.5) | 82 (29.8) | <0.001 |
| GLP-1 analogs | 0 (0) | 4 (1.5) | 0.124 |
| Insulin | 120 (43.6) | 91 (33.1) | 0.011 |
| Antihypertensive medications | | | |
| Yes | 261 (94.9) | 191 (69.5) | <0.001 |
| No | 14 (5.1) | 84 (30.5) | |
| ARB | 183 (66.8) | 145 (52.7) | 0.001 |
| ACE inhibitors | 57 (20.7) | 10 (3.6) | <0.001 |
| CCB | 66 (24.0) | 99 (36.0) | 0.002 |
| Beta-blockers | 156 (56.7) | 61 (22.2) | <0.001 |
| Diuretics | 163 (59.3) | 30 (10.9) | <0.001 |
| Aspirin use | | | |
| Yes | 272 (98.9) | 32 (11.6) | <0.001 |
| No | 3 (1.1) | 243 (88.4) | |
| Fibrate use | | | |
| Yes | 7 (2.5) | 22 (8.0) | 0.004 |
| No | 268 (97.5) | 253 (92.0) | |
| Statin use | | | |
| Yes | 270 (98.2) | 214 (77.8) | <0.001 |
| No | 5 (1.8) | 61 (22.2) | |

case group (8.7%) than in the control group (12.9%, p = 0.037). Figure 1 illustrates the linkage disequilibrium between allele pairs of the three SNPs. High linkage disequilibrium was observed between rs266729 and rs2241766, as well as between rs2241766 and rs1501299.

Table S1 presents Hardy-Weinberg Equilibrium (HWE) for three *ADIPOQ* gene polymorphisms (rs266729, rs2241766, rs1501299) in total as well as in case, and control

Table 3 Genotype, allele, distribution, and haplotype of ADIPOQ SNPs.

| Characteristic | Groups | | p-value |
|---|---|---|---|
| | Case (n = 275) | Control (n = 275) | |
| **rs266729** | | | |
| C/C | 150 (54.5) | 140 (50.9) | 0.058 |
| C/G | 111 (40.4) | 106 (38.5) | |
| G/G | 14 (5.1) | 29 (10.5) | |
| **rs2241766** | | | |
| T/T | 123 (44.7) | 166 (60.4) | <0.001 |
| T/G | 124 (45.1) | 81 (29.5) | |
| G/G | 28 (10.2) | 28 (10.2) | |
| **rs1501299** | | | |
| G/G | 165 (60.0) | 148 (53.8) | 0.069 |
| G/T | 97 (35.3) | 101 (36.7) | |
| T/T | 13 (4.7) | 26 (9.5) | |
| **Allele rs266729** (n = 1100) | | | |
| C | 411 (74.7) | 386 (70.2) | 0.092 |
| G | 139 (25.3) | 164 (29.8) | |
| **Allele rs2241766** (n = 1100) | | | |
| T | 370 (67.3) | 413 (75.1) | 0.004 |
| G | 180 (32.7) | 137 (24.9) | |
| **Allele rs1501299** (n = 1100) | | | |
| G | 427 (77.6) | 397 (72.2) | 0.037 |
| T | 123 (22.4) | 153 (27.8) | |
| **Haplotype rs266729 + rs2241766 + rs1501299** (n = 1100) | | | |
| C-T-G | 240 (43.6) | 230 (41.8) | 0.028 |
| C-G-G | 113 (20.5) | 85 (15.5) | |
| G-T-T | 48 (8.7) | 71 (12.9) | |
| G-T-G | 42 (7.6) | 58 (10.5) | |
| C-T-T | 40 (7.3) | 54 (9.8) | |
| G-G-G | 32 (5.8) | 24 (4.4) | |
| C-G-T | 18 (3.3) | 17 (3.1) | |
| G-G-T | 17 (3.1) | 11 (2.0) | |

groups. All were in HWE across these groups, except rs2241766, which significantly deviated in the total and control groups but remained in HWE in the case group.

Table 4 shows the results of univariate and multivariate logistic regression analyses evaluating the association between *ADIPOQ* gene variants and AMI in T2DM patients. The G/G genotype of rs266729 was associated with a lower risk of AMI compared to the C/C genotype (OR = 0.45, 95% CI [0.23–0.89], $p$ = 0.021) in univariate analysis, but the association was not statistically significant in multivariate analysis (OR = 0.58, 95% CI [0.27–1.27], $p$ = 0.174). For rs2241766, patients with the T/G genotype had a significantly

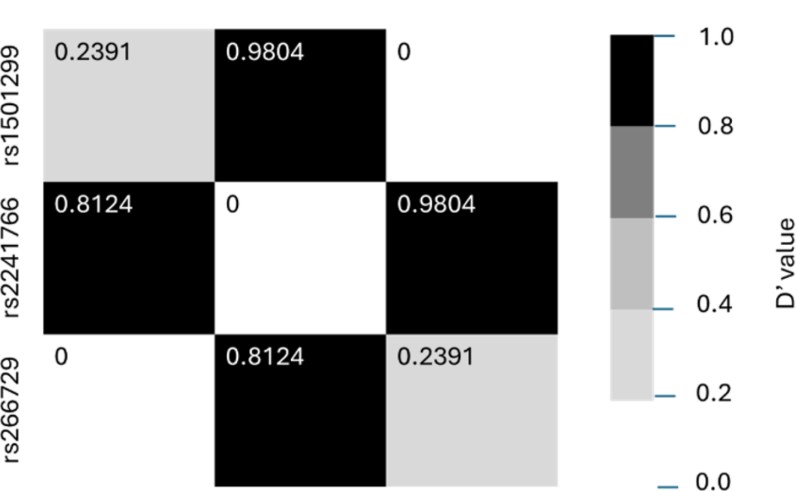

**Figure 1 Linkage disequilibrium between allele pairs of rs266729, rs2241766 and rs1501299.**

| Characteristic | Crude | | Adjusted[a] | |
|---|---|---|---|---|
| | OR (KTC 95%) | $p$ | OR (KTC 95%) | $p$ |
| **rs266729** | | | | |
| C/C | 1 | | 1 | |
| C/G | 0.98 (0.69–1.39) | 0.899 | 1.21 (0.81–1.81) | 0.360 |
| G/G | 0.45 (0.23–0.89) | 0.021 | 0.58 (0.27–1.27) | 0.174 |
| **rs2241766** | | | | |
| T/T | 1 | | 1 | |
| T/G | 2.07 (1.44–2.97) | <0.001 | 1.87 (1.23–2.84) | 0.004 |
| G/G | 1.35 (0.76–2.39) | 0.305 | 1.55 (0.80–3.02) | 0.197 |
| **rs1501299** | | | | |
| G/G | 1 | | 1 | |
| G/T | 0.86 (0.60–1.23) | 0.412 | 0.86 (0.57–1.29) | 0.457 |
| T/T | 0.45 (0.22–0.90) | 0.025 | 0.49 (0.22–1.10) | 0.083 |
| **Allele rs266729** ($n = 1100$) | | | | |
| C | 1 | | 1 | |
| G | 0.80 (0.61–1.04) | 0.092 | 0.95 (0.70–1.29) | 0.736 |
| **Allele rs2241766** ($n = 1100$) | | | | |
| T | 1 | | 1 | |
| G | 1.47 (1.13–1.91) | 0.004 | 1.47 (1.09–1.98) | 0.013 |
| **Allele rs1501299** ($n = 1100$) | | | | |
| G | 1 | | 1 | |
| T | 0.75 (0.57–0.98) | 0.037 | 0.77 (0.56–1.05) | 0.103 |

Table 4 Association between ADIPOQ gene variants and acute myocardial infarction.

(Continued)

| Characteristic | Crude | | Adjusted[a] | |
|---|---|---|---|---|
| | OR (KTC 95%) | *p* | OR (KTC 95%) | *p* |
| **Allele rs266729 + rs2241766 + rs1501299 (*n* = 1100)** | | | | |
| C-T-G | 1 | | 1 | |
| C-G-G | 1.27 (0.91–1.78) | 0.156 | 1.37 (0.93–2.01) | 0.111 |
| G-T-T | 0.65 (0.43–0.97) | 0.037 | 0.92 (0.57–1.48) | 0.732 |
| G-T-G | 0.69 (0.45–1.07) | 0.101 | 0.71 (0.43–1.17) | 0.176 |
| C-T-T | 0.71 (0.45–1.11) | 0.133 | 0.57 (0.35–0.95) | 0.030 |
| G-G-G | 1.28 (0.73–2.24) | 0.390 | 1.30 (0.70–2.43) | 0.404 |
| C-G-T | 1.01 (0.51–2.02) | 0.967 | 0.85 (0.40–1.80) | 0.670 |
| G-G-T | 1.48 (0.68–3.23) | 0.323 | 1.65 (0.62–4.43) | 0.319 |

**Note:**
[a] Adjusted for age, gender, diabetes duration, hypertension, BMI, smoking status, HbA1c, total cholesterol, HDL-C, LDL-C, and triglycerides

higher risk of AMI compared to those with the T/T genotype (OR = 2.07, 95% CI [1.44–2.97], *p* < 0.001 in univariate analysis; OR = 1.87, 95% CI [1.23–2.84], *p* = 0.004 in multivariate analysis). The T/T genotype of rs1501299 was associated with a reduced risk of AMI in univariate analysis (OR = 0.45, 95% CI [0.22–0.90], *p* = 0.025), but this association was not statistically significant in multivariate analysis (OR = 0.49, 95% CI [0.22–1.10], *p* = 0.083). Haplotype analysis showed that the G-T-T haplotype was associated with a lower risk of AMI compared to the C-T-G haplotype (OR = 0.65, 95% CI [0.43–0.97], *p* = 0.037), though this result lost significance in multivariate analysis (OR = 0.92, 95% CI [0.57–1.48], *p* = 0.732). Conversely, the C-T-T haplotype was significantly associated with a reduced risk of AMI in multivariate analysis (OR = 0.57, 95% CI [0.35–0.95], *p* = 0.030).

## DISCUSSION

While studies examining the effects of *ADIPOQ* gene variants have been conducted globally, showing differences across various populations, our study is one of the first to explore this topic in Vietnam. Our research provides additional genetic data from a Southeast Asian country, further confirming the role and potential of *ADIPOQ* gene variants in predicting AMI among T2DM patients. The analysis revealed significant differences in genotype distribution between the case and control groups. Specifically, the rs266729 and rs1501299 variants were shown to be associated with a decreased risk of AMI, while the rs2241766 variant was associated with an increased risk.

### Protective variants: rs266729

The rs266729 (−11,377 C/G) variant is a SNP located in the promoter region of the *ADIPOQ* gene on chromosome 3q27. This position directly influences the expression of adiponectin, a key hormone in metabolic and vascular homeostasis. The G allele is linked to higher adiponectin expression, which confers cardioprotection through multiple downstream pathways, including enhanced insulin sensitivity *via* 5′-Adenosine

monophosphate-activated protein kinase (AMPK) activation in skeletal muscle and liver, suppression of pro-inflammatory cytokines such as TNF-α and IL-6, and improved endothelial function through increased nitric oxide production and inhibition of endothelial adhesion molecule expression.

In terms of distribution, among T2DM patients with AMI, the frequency of the C/C genotype ranged from 48.1% to 60.3% (*Zhang et al., 2018*; *Cheung et al., 2014*), with our study finding it at 54.5%, similar to studies conducted in France (*Lacquemant et al., 2004*) and China (*Zhong et al., 2010*). The frequencies of the C/G and G/G genotypes in our study were 40.4% and 5.1%, respectively, which align with studies in the UK, Japan, and the USA (*Gable et al., 2007*; *Hegener et al., 2006*; *Oguri et al., 2009*). The allele frequencies of C and G in our study were 74.7% and 25.3%, respectively, which are within the range of international studies (*Lacquemant et al., 2004*; *Hegener et al., 2006*; *Chiodini et al., 2010*). Among T2DM patients without AMI, the frequency of the C/C genotype in our study was 50.9%, lower than in European studies (*Chiodini et al., 2010*) but similar to that in China (*Zhang et al., 2018*). The G/G genotype frequency in our study (10.5%) was unusually high compared to other populations, particularly in Europe (ranging from 4.4% to 7.0%) (*Lacquemant et al., 2004*; *Gable et al., 2007*; *Chiodini et al., 2010*). The G allele frequency in our study (29.8%) was also higher than in most European countries and Japan (*Lacquemant et al., 2004*; *Gable et al., 2007*; *Oguri et al., 2009*; *Chiodini et al., 2010*).

Our study found that the G/G genotype of rs266729 was associated with a reduced risk of AMI, consistent with the study by *Qi et al. (2006)*, which linked the G allele to higher adiponectin levels, thereby reducing the risk of CAD. Similarly, *Hegener et al. (2006)* also found that rs266729 was associated with a reduced risk of ischemic stroke. The consistency between our study and *Qi et al. (2006)* could be due to the focus on patients with type 2 diabetes, a clear risk factor for CAD, making it easier to detect the protective association of rs266729. In contrast, studies by *Lacquemant et al. (2004)*, *Gable et al. (2007)*, and *Antonopoulos et al. (2013)* did not find significant associations between rs266729 and CAD risk. European studies often involve more diverse populations, not focusing on a specific condition like diabetes but including other risk factors such as obesity and metabolic syndrome. The multi-center study design could also lead to heterogeneity in data analysis, which may explain unclear association between rs266729 and cardiovascular disease in these studies.

## Protective variants: rs1501299

The rs1501299 (+276 G/T) variant is an intronic SNP within the *ADIPOQ* gene that has been widely associated with metabolic conditions, including T2DM and CAD. Although intronic, this SNP is postulated to be in a key regulatory region, influencing *ADIPOQ* transcription by altering enhancer or silencer activity. Consequently, the T allele is linked to higher transcriptional output and increased circulating adiponectin, whose protective effects are mediated *via* its well-established anti-inflammatory, anti-atherogenic, and insulin-sensitizing properties.

In terms of genotype distribution, among T2DM patients with AMI, the frequency of the G/G genotype in our study was 60.0%, higher than in most European and American

studies (*Antonopoulos et al., 2013*; *Lacquemant et al., 2004*; *Gable et al., 2007*; *Hegener et al., 2006*; *Chiodini et al., 2010*) but similar to findings in Japan (60.6%) (*Katakami et al., 2012*). The G/T genotype frequency was 35.3%, comparable to studies in Italy (37.9%) (*Chiodini et al., 2010*) and the USA (37.1%) (*Qi et al., 2006*). The T/T genotype frequency in our study was 4.7%, higher than in China (1.1%) (*Zhang et al., 2018*) but lower than in some Middle Eastern populations such as Qatar (14.0%) (*Rizk et al., 2013*). Among T2DM patients without AMI, the frequency of the G/G genotype in Vietnam was 53.8%, close to that found in the USA (53.1%) (*Hegener et al., 2006*) and Hong Kong (54.9%) (*Cheung et al., 2014*). The G/T genotype frequency was 36.7%, similar to that in Italy (*Bacci et al., 2004*) and the USA (*Qi et al., 2006*). The T/T genotype frequency was 9.5%, higher than in China (1.6%) (*Zhang et al., 2018*) but lower than in Iran (13.4%) (*Esteghamati et al., 2012*). The G allele frequency in our study was 72.2%, aligning with studies from Hong Kong (73.8%) (*Cheung et al., 2014*) and Poland (74.2%) (*Ambroziak et al., 2018*).

In our study, we found that the T/T genotype was associated with a reduced risk of AMI, supporting the protective role of adiponectin in reducing inflammation and improving insulin sensitivity. This finding is consistent with the studies by *Bacci et al. (2004)* and *Qi et al. (2006)*, which linked the T/T genotype to higher adiponectin levels and a reduced risk of CAD. *Hegener et al. (2006)* also identified an association between rs1501299 and a lower risk of vascular diseases, although it did not reach statistical significance for AMI. In contrast, studies by *Zhong et al. (2010)* in China, *Antonopoulos et al. (2013)* in Greece, and *Oguri et al. (2009)* in Japan did not find significant associations between rs1501299 and CAD risk. The differences in genetic structures between the Chinese, Greek, and Japanese populations compared to the Vietnamese and European populations might influence the effect of rs1501299 on CAD risk. Moreover, *Zhong et al. (2010)* and *Oguri et al. (2009)*, despite having large sample sizes, did not focus on T2DM patients, which could explain the lack of a clear genetic association, as found in our study focusing on T2DM patients. *Antonopoulos et al. (2013)* emphasized endothelial function, highlighting the complex relationship between *ADIPOQ* gene variants and CAD, which may be influenced by intermediary factors like endothelial dysfunction.

## Risk-associated variant: rs2241766

The rs2241766 (+45 T/G) variant is a missense SNP located in exon 2 of the *ADIPOQ* gene that results in a threonine-to-glycine substitution. This amino acid change is postulated to disrupt the post-translational processing and multimerization of adiponectin into its most biologically active high-molecular-weight (HMW) forms. Consequently, the G allele is associated with lower circulating levels of functional adiponectin, which can foster a pro-inflammatory and pro-atherogenic state, thereby increasing AMI risk.

In terms of distribution, among T2DM patients with AMI, the frequency of the T/T genotype in our study was 44.7%, significantly lower than in Europe (*Antonopoulos et al., 2013*; *Lacquemant et al., 2004*; *Gable et al., 2007*; *Chiodini et al., 2010*) and the USA (*Hegener et al., 2006*; *Qi et al., 2006*; *Qi et al., 2005*), but comparable to studies from the Middle East and Asia, such as Qatar (43.7%) (*Rizk et al., 2013*) and Iran (42.1%) (*Esteghamati et al., 2012*). The frequency of the T/G genotype in our study was 45.1%,
relatively high compared to other countries (*Antonopoulos et al., 2013*; *Lacquemant et al., 2004*; *Hegener et al., 2006*; *Bacci et al., 2004*; *Shaker & Ismail, 2014*), and the same as in Hong Kong (45.1%) (*Cheung et al., 2014*). The G/G genotype frequency was 10.2%, higher than in most studies from Europe (*Antonopoulos et al., 2013*; *Lacquemant et al., 2004*; *Gable et al., 2007*; *Chiodini et al., 2010*; *Bacci et al., 2004*) and the USA (*Hegener et al., 2006*). Among T2DM patients without AMI, the frequency of the T/T genotype in our study (60.4%) was close to South Korea (56.3%) (*Ji et al., 2018*) but significantly lower than in Africa (*e.g.*, Egypt 93.3% (*Shaker & Ismail, 2014*)). The frequency of the T/G genotype was 29.5%, similar to studies in the UK (*Gable et al., 2007*) and Italy (*Bacci et al., 2004*), but higher than in Africa (*Shaker & Ismail, 2014*; *Saleh et al., 2020*). The G/G genotype frequency in our study (10.2%) was much higher than in Europe (*Antonopoulos et al., 2013*; *Lacquemant et al., 2004*; *Gable et al., 2007*; *Chiodini et al., 2010*; *Bacci et al., 2004*) and the USA (*Hegener et al., 2006*) but still lower than some studies from the Middle East (*Rizk et al., 2013*).

The G allele of the missense variant rs2241766 was associated with an increased risk of AMI. This observation is in agreement with findings from other high-risk T2DM populations, *Ji et al. (2018)*, *Lacquemant et al. (2004)*, *Shaker & Ismail (2014)* suggesting a potentially consistent detrimental role for this variant across diverse ethnicities. Meanwhile, *Hegener et al. (2006)* found no significant association between rs2241766 and the risk of myocardial infarction or stroke. It is important to note that *Hegener et al. (2006)* was conducted on a group of healthy males with a low prevalence of T2DM, which may explain the lack of association between rs2241766 and myocardial infarction risk. This suggests that the link between rs2241766 and CAD may be more pronounced in high-risk groups, such as T2DM patients.

Hardy-Weinberg equilibrium (HWE) was assessed as a standard quality control measure for all SNPs. The rs266729 and rs1501299 variants conformed to HWE expectations, indicating robust genotyping data. However, a significant deviation from HWE was observed for rs2241766, primarily driven by the control group ($p = 0.00063$). While genotyping error is less likely, as other SNPs were in equilibrium, this deviation may reflect subtle population stratification or an inadvertent selection bias in our hospital-based control group.

Crucially, the association between rs2241766 and increased AMI risk remained highly significant after multivariate adjustment (OR = 1.87, 95% CI [1.23–2.84], $p = 0.004$), suggesting that this is a robust biological finding rather than a statistical artifact. Nevertheless, we acknowledge this limitation and recommend that future studies utilize population-based controls or apply methods to correct for potential stratification to further validate these findings.

## Haplotype analysis

Several studies have evaluated haplotype analysis to simultaneously determine the role of multiple SNPs in understanding their association with specific diseases, such as AMI. Lu *Qi et al.*'s *(2006)* study indicated that the T-T interaction of two SNPs, rs2241766 and rs1501299, was associated with a reduced risk of cardiovascular disease compared to the

T-G interaction in women with type 2 diabetes in the U.S. *Gable et al.*'s *(2007)* study showed that the T-T and G-G interactions of these two SNPs had different effects on the risk of MI in White populations, although the differences were not statistically significant. *Zhang et al.*'s *(2018)* study found that the interaction between three SNPs, rs266729, rs2241766, and rs1501299, in the C-G-G and C-T-G configurations could increase myocardial infarction risk compared to the C-T-T interaction in the Han Chinese population. Our study (2024) discovered that the interaction between the three SNPs, rs266729, rs2241766, and rs1501299, in the G-T-T configuration may reduce myocardial infarction risk compared to the C-T-G configuration in type 2 diabetes patients. These studies collectively suggest that interactions between *ADIPOQ* gene variants may have significant associations with AMI risk, particularly the T-T interaction of rs2241766 and rs1501299, which is linked to a reduced risk of AMI.

### Limitations

Our study has several limitations. First, although the research focuses on variants within the *ADIPOQ* gene, other genetic factors that could influence AMI risk were not analyzed. Additionally, environmental factors such as diet, physical activity, and socioeconomic status, which may affect AMI risk, were not considered. To properly assess the impact of *ADIPOQ* gene variants on AMI, these competing or confounding risk factors should be more thoroughly examined. Second, the study is limited to T2DM patients from a single center, lacking representation for the broader Vietnamese population, which may restrict the comparability with international studies and the generalizability of the results to other populations. Since the distribution of *ADIPOQ* gene variants and its association with AMI may vary across different ethnic groups, and Vietnam itself has 54 ethnic groups, more studies like this are needed in various populations. Third, in this study, we assumed that *ADIPOQ* gene variants influence adiponectin levels, thereby affecting AMI risk. However, we did not directly assess how these SNPs specifically affect adiponectin levels or how such changes in adiponectin impact AMI risk. Therefore, our study is limited in explaining the biological mechanisms and in comparing the findings with other studies. Future research should focus on investigating the mechanisms and specific effects of *ADIPOQ* gene variants on AMI. Finally, due to the retrospective nature of this case-control study, many variables were assessed based solely on existing medical records. As such, this approach may introduce certain limitations in fully understanding how specific variables were defined or documented. For example, the individual contributions of the 2018 AMI diagnostic sub-criteria (*e.g.*, ischemic symptoms, ECG changes, echocardiographic findings, angiographic evidence) were not recorded separately in our dataset. Future prospective studies should consider systematically documenting these components to enable more detailed and transparent analysis.

## CONCLUSIONS

Our study found that rs266729 (G/G) and rs1501299 (T/T) are associated with the reduced risk of myocardial infarction. In contrast, rs2241766 (T/G) doubles the risk of myocardial

infarction. Future studies should focus on quantifying adiponectin levels, exploring other gene variants, and evaluating gene-environment interactions. Such studies will help clarify the physiological mechanisms of these variants, strengthen the reliability of research findings, and provide a more complete picture of cardiovascular risk in Vietnamese T2DM patients.

## ACKNOWLEDGEMENTS

We thank all patients for their participation in this study, and the Director Board of University Medical Center at Ho Chi Minh City for their continuous support during our data collection.

### Funding
The authors received no funding for this work.

### Competing Interests
Truc Thanh Thai is an Academic Editor for PeerJ.

### Author Contributions
- Bao Hoang Le conceived and designed the experiments, performed the experiments, prepared figures and/or tables, authored or reviewed drafts of the article, and approved the final draft.
- Khanh Quang Tran conceived and designed the experiments, performed the experiments, authored or reviewed drafts of the article, and approved the final draft.
- Nhu Nhat Quynh Nguyen performed the experiments, authored or reviewed drafts of the article, and approved the final draft.
- Truc Thanh Thai analyzed the data, prepared figures and/or tables, authored or reviewed drafts of the article, and approved the final draft.

### Human Ethics
The following information was supplied relating to ethical approvals (*i.e.*, approving body and any reference numbers):

Biomedical Research Ethics Committee of the University of Medicine and Pharmacy at Ho Chi Minh City

### Data Availability
The raw data and data dictionary are available in the Supplemental Files.

### Supplemental Information
Supplemental information for this article can be found online at http://dx.doi.org/10.7717/peerj.20145#supplemental-information.

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
