# Peer review of "The association between ADIPOQ gene variants (rs266729, rs2241766, rs1501299) and acute myocardial infarction in Vietnamese patients with type 2 diabetes mellitus"

_PeerJ, doi:10.7717/peerj.20145_

## Round 0.1 · original submission · Major Revisions

Reviewer 1 ·

Basic reporting

The authors provide sufficient context for the subject, and the figures are relevant,clear and comprehensible. However, there are some errors throughout the manuscript that need editing.

Experimental design

In this section, the authors touched on the design of the study and the region, but it was need some editing.

Validity of the findings

The data is abundantly available as it is robust and statistically sound. Conclusions are clear, the study offers sample data for the authors to draw, but the data needs more care and linking the results together to obtain more objective, more important and more useful results.

Additional comments

(2) Changes which must be made before publication

• In this study, the association between rs266729, rs2241766, and rs1501299 polymorphisms in the ADIPOQ gene were studied and acute myocardial infarction risk in patients with type 2 diabetes mellitus (T2DM) in Vietnam.
• Title: the title properly explains the purpose and objective of the article.

Abstract:
• The abstract offers an accurate summary of the paper, and the language used in the abstract is easy to read and understand.
• Line 26-28 : I suggested change to " This study looks at how the rs266729, rs2241766, and rs1501299 variations in the ADIPOQ gene are related to the risk of acute myocardial infarction in patients with type 2 diabetes in Vietnam.
• Line 31-32: Please rephrase the sentence for more clarity (Patients were matched based on age, gender, smoking status, and duration of T2DM)because we did not touch on this thing in the results within the research.
• Line 34-36: It is repetitive and has no effect on the results. Please I suggest it is better to cancel it.
Keywords:
• Line 46-47: The key words should be differ: gene polymorphism to polymorphisms , also delete adiponectin, Vietnam to another word if you want .

Introduction:
• Introduction: The authors provide sufficient context on the subject. but please keep it brief. The introduction is very long and contains unnecessary verbiage.
• Line 51-75: Please rephrase the paragraph briefly for more clarity.
• Line 67: Please use "increasing "instead "increase"
• Line 77: Please use "the Chinese "instead "Chinese"
• Line 58-86: Please rephrase the paragraph briefly for more clarity.
• Materials and methods: In this section, the authors touched on the design of the study and the region, but I suggest that this section need some changes such as:
• Line 99: Please use "in Ho Chi "instead "at Ho Chi "
• Line 99-102: Please delete the paragraph from the beginning of the sentence " Each year, ……….. to corresponding departments", I think there is no need to prolong it.
• Line 107 : Age must be specified and a specific rate must be given from to, not from 18 and above, please change it.
• Line 108: Please use " AMI and treated" instead " AMI, treated "
• Line 114-117 : Please delete the paragraph from the beginning of the sentence "The input parameters ………. to 80% 117 power)".
• Line 129-130: This is a result, not a working method. It should be moved to the results or deleted.
• Line 146: Please use "the BigDye® "instead " BigDye® "
Results: The findings are presented clearly.
• Line 173: Please use " sociodemographic " instead " socio-demographic "
• Lines 173-190: I think there is no need for Tables 1 and 2, as what is in them are known facts that have been studied previously and did not add anything to knowledge, but if the factors that have a significant impact such as (glucose , HbA1c , total cholesterol, LDL-C, triglycerides, HDL-C and estimated glomerular filtration rate (eGFR) were linked with the genetic variables in Table No. 3, perhaps there would be a certain importance, and thus their presence would be merely an elongation of the research. I suggest either linking the tables or removing them, as it is better to mention the rates of these factors in general as numbers within the method of work to show the case only. Please rephrase the sentence for more clarity.
Discussion: A bit lengthy, it needs to be shortened as much as possible.
• Line 229: Please use " risk." instead " the risk. "
• Line 342: Please use " affects " instead " affect "
Conclusions are clear, the study offers sample data for the authors to draw.
• Line 347 : Please use " reduced " instead " reduce "
Grammar: is good along the manuscriptو but there are some corrections mentioned above.

Annotated reviews are not available for download in order to protect the identity of reviewers who chose to remain anonymous.

Reviewer 2 ·

Basic reporting

Comment 1: Line 101: Reframe the statement
Comment 2: Line 102: Reframe the statement
Comment 3: Line 107 : Reframe the statement
Comment 4: Line 348: ….associated with reduce risk …. grammatical Suggestion: ..reduced.

Experimental design

Comment 5: Mention the minimum and maximum age group of the patients
Comment 6: Mention the mean duration of T2DM for the onset of AMI
Comment 7: Kindly mention the number of AMI patients falling into each of the AMI 2018 critera
Comment 8: SNPs wild type allele/variant allele for the studied polymorphisms are not mentioned anywhere in manuscript
Comment 9: Line 67: Changes in these polymorphisms SNPs not mentioned kindly mention the SNPs included in the reference
Comment 10: Line 277: rs2241766 had a protective effect.. which allele or genotype..not mentioned
Comment 11: In Table 2 & 3 controls in column not mentioned, both are mentioned as cases kindly correct it

Validity of the findings

Comment 12: Although your results are compelling, the data analysis should be improved by performing multi variate analysis on Socio-demographic and clinical characteristics with respect to the studied SNPs
Comment 13: Include detailed discussion
Comment 14: Mention Hardy-Weinberg equilibrium in the tables
Comment 15: Explain in detail the importance of ADIPOQ gene in T2DM and AMI

Additional comments

I commend the authors a systematic review and meta analysis was conducted for the inclusion of the subjects under study. The manuscript can be strengthened with the given comments to increase its valuable contributions to cardiovascular genetics in a T2DM cohort and to reach the standards of the journal

---

## Round 0.2 · Minor Revisions

Dear authors,

Thank you for your submission and for addressing all the reviewers' comments. Before moving to production, we need some clarifications to be incorporated into your manuscript (as needed), ensuring methodological transparency and reproducibility:

- instrument details and settings (provide full details - model, manufacturer - of all critical equipment, especially analytical instruments, including specs, centrifuges, balances.... Induce also specific parameters where applicable such as wavelengths, slit widths, currents, background correction mode, etc), Ie sufficient detail for replication

- reagents and standards (please include catalog numbers / #ref, suppliers, and purity grades for all key reagents, including chemicals, kits, and calibration standards; and explicitly describe preparation of standards, eg concentration ranges, matrix modifiers if used)

- QC and replication (describe whether replicate measurements were performed explicitly - technical and/or biological replicates - where relevant; clarify if blank runs, duplicate assays, or certified reference materials were used for accuracy checks; and state when possible detection limits, calibration verification steps, and any measures taken to address matrix effects)

- sample handling (make sure that all important details on storage conditions are disclosed, namely specific storage conditions - temperature, duration - if freeze-thaw cycles occured, how many? & note how they were minimized or controlled
- statistics (while software packages were listed, indicate how assumptions were checked eg normality. Were made effect size assumptions, alpha level, target power?. & Explicit inclusion / exclusion criteria should be reported in the methods, not just tables)

- any deposition of raw or processed data in public repositories (I am sorry if I missed the disclosure)

Thank you.

Reviewer 1 ·

Basic reporting

I would like to thank the authors for carefully addressing all the modifications requested by the reviewers.

Experimental design

In this section, the authors touched on the design of the study and the region, and all edits by the author have been well done.

Validity of the findings

The data are abundant, due to their accuracy and statistical validity. The conclusions are clear, the study provides representative data for the authors, and the data that needed modification has been modified.

Reviewer 2 ·

Basic reporting

Comment 1: Line 101: Reframe the statement
Comment 2: Line 102: Reframe the statement
Comment 3: Line 107: Reframe the statement
Comment 4: Line 348: ….associated with reduce risk …. grammatical Suggestion: ..reduced.

The author made all the corrections and suggestions.

Experimental design

Comment 5: Mention the minimum and maximum age groups of the patients
Comment 6: Mention the mean duration of T2DM for the onset of AMI
Comment 7: Kindly mention the number of AMI patients falling into each of the AMI 2018 criteria
Comment 8: SNPs wild-type allele/variant allele for the studied polymorphisms are not mentioned anywhere in the manuscript
Comment 9: Line 67: Changes in these polymorphism SNPs, not mentioned, kindly mention the SNPs included in the
reference
Comment 10: Line 277: rs2241766 had a protective effect. Which allele or genotype..not mentioned
Comment 11: In Tables 2 and 3, controls in the column not mentioned are mentioned as cases. Kindly correct it

The author made all the corrections and suggestions.

Validity of the findings

Comment 12: Although your results are compelling, the data analysis should be improved by performing multivariate analysis on Socio-demographic and clinical characteristics with respect to the studied SNPs
Comment 13: Include detailed discussion
Comment 14: Mention Hardy-Weinberg equilibrium in the tables
Comment 15: Explain in detail the importance of the ADIPOQ gene in T2DM and AMI

The author made all the corrections and suggestions.

Additional comments

We have reviewed the revised manuscript and confirm that all the suggested corrections have been addressed satisfactorily. The authors have incorporated the necessary changes, and the content now meets the required standards for clarity, scientific accuracy, and completeness.

---

## Round 0.3 · accepted · Accept

Dear authors,

Many thanks for your work and revisions. I am now accepting your manuscript for publication in PeerJ. Congratulations and thank you once again.